# Neutrophil–Lymphocyte and Platelet–Lymphocyte Ratios in Preoperative Differential Diagnosis of Benign, Borderline, and Malignant Ovarian Tumors

**DOI:** 10.3390/jcm11051355

**Published:** 2022-03-01

**Authors:** Tae Hui Yun, Yoon Young Jeong, Sun Jae Lee, Youn Seok Choi, Jung Min Ryu

**Affiliations:** 1Department of Obstetrics and Gynecology, School of Medicine, Daegu Catholic University, Daegu 42472, Korea; tahh0123@naver.com (T.H.Y.); nning91@naver.com (Y.Y.J.); 2Department of Pathology, School of Medicine, Daegu Catholic University, Daegu 42472, Korea; pathosjlee@cu.ac.kr

**Keywords:** ovarian neoplasm, neutrophils, lymphocytes, platelet count

## Abstract

The purpose of this study was to investigate whether the neutrophil–lymphocyte ratio (NLR) and platelet–lymphocyte ratio (PLR) can be used as supplementary tools to differentiate between benign, borderline, and malignant ovarian tumors. The ratio of patients with benign to borderline to malignant tumors was planned as 3:1:2 considering the incidence of each disease. Consecutive patients were enrolled retrospectively. Preoperative complete blood counts with differentials were investigated, and calculated NLRs and PLRs were analyzed. A total of 630 patients with ovarian tumors were enrolled in this study. The final histopathological results revealed that 318 patients had benign, 108 patients had epithelial borderline, and 204 patients had epithelial malignant ovarian tumors. The NLR and PLR were significantly higher in malignant than in benign or borderline ovarian tumors, and they did not differ significantly between benign and borderline ovarian tumors. The diagnostic cut-off value of NLR for differentiating between benign or borderline and malignant tumors was 2.36, whereas that of PLR for differentiating between benign/borderline and malignancy was 150.02. High preoperative NLR and PLR indicate that the likelihood of epithelial ovarian cancer is higher than that of benign or borderline tumors.

## 1. Introduction

Most cases of epithelial ovarian cancer are asymptomatic in the early stages, and there is currently no adequate screening test for early diagnosis, so they are often detected at an advanced stage and have a poor prognosis [1]. Borderline ovarian tumors show excellent prognosis because the rates of metastasis and recurrence are low, and most patients are detected at an early stage and can be cured by surgical treatment. Preoperative biopsy is not recommended due to the risk of spillage of tumor cells in the abdominal cavity, so the diagnosis is confirmed by histopathologic findings after surgery. Preoperative diagnosis of ovarian tumors mainly depends on imaging studies including ultrasound and CT [2,3]. However, it is not always easy to differentially diagnose benign, borderline, and malignant ovarian tumors only by imaging findings. Tumor markers such as CA125 and CA19-9 also play an adjunctive role in diagnosing of ovarian tumors, but are not diagnostic because of their low specificity [4].

Thrombocytosis can be observed in tumor formation and oncogenesis [5]. A recent study reported that thrombocytosis is associated with an undiagnosed cancer, and with a 7.11-fold relative risk particularly for ovarian cancer [6]. In addition, several studies have shown that hematologic findings such as the neutrophil–lymphocyte ratio (NLR) and platelet–lymphocyte ratio (PLR) are useful as a supplementary role in the differential diagnosis of ovarian tumors [7,8,9,10,11,12,13,14,15]. These studies reported that NLR and PLR levels tend to increase in malignant ovarian tumors. In addition, it has been reported that an increase in NLR and PLR is associated with poor prognosis in cancer patients [16,17,18,19]. However, NLR and PLR increases are not cancer-specific, and such increases can also be observed in systemic diseases such as cardiovascular disease, rheumatic disease, and infectious disease [20,21,22,23].

Most of the studies on NLR and/or PLR of ovarian tumor patients are about the difference between benign and malignant tumors, and there are only a few studies on borderline ovarian tumors [12,13,24]. There were two studies that investigated both the NLR and PLR of patients with borderline ovarian tumors and compared them with benign and malignant tumors. Those studies showed that NLR and PLR showed higher levels in patients with malignant tumors than in patients with benign tumors, but conflicting results in borderline ovarian tumors. One study reported that NLR and PLR levels in patients with borderline ovarian tumors were similar to those of benign tumors [13], and the other study reported that the levels were similar to those of malignant tumors [24].

The purpose of this study was to investigate the differences in NLR and PLR levels in patients with borderline ovarian tumor compared to patients with benign and malignant ovarian tumors, and to determine whether they can be used for preoperative differential diagnosis.

## 2. Materials and Methods

In this study, all patients with ovarian tumors identified by preoperative imaging studies such as ultrasonography, CT or MRI were included. Patients who underwent surgery at Daegu Catholic University Hospital and were diagnosed with benign ovarian tumors (epithelial (serous, mucinous, seromucinous, etc.), non-epithelial (mature cystic teratoma, fibroma, thecoma, etc.)), borderline epithelial ovarian tumors, or malignant epithelial ovarian tumors upon histological examination were included in the present study.

It is difficult to accurately estimate the incidence of ovarian cyst according to each histological classification [25,26,27]. Certain ovarian cysts are functional and usually do not require surgery. Therefore, the sample size was determined in the order of benign, malignant, and borderline in consideration of the order of incidence of ovarian tumors. The sample size was calculated using the MedCalc Statistical Software version 19.4.0 (MedCalc Software Ltd., Ostend, Belgium, 2020) with reference to the NLR values of the results of a previous study by Polat et al. [13]. The required sample size was calculated as 305 for the benign ovarian tumor group and 204 for the malignant group according to the following conditions: difference of mean of two group = 0.9, standard deviation in benign group = 2.9, standard deviation in malignant group = 3.9, ratio of sample size in benign/malignant ovarian tumor group = 1.5, statistical power (1-β) 80%, and significance level (α) 0.05 (two-sided test).

Consecutive patients were enrolled in the present study retrospectively such that patients with each disease met the following criteria: benign (*n* > 300, from September 2010 to July 2021), borderline (*n* > 100, from December 2006 to July 2021), and malignant (*n* > 200, from November 2002 to July 2021). According to the final pathological report, the patients were divided into benign, borderline, and malignant ovarian tumor groups. All histopathological results from ovarian tumor specimens were reviewed by an expert gynecologic pathologist (Lee, S.J., one of the authors of this study).

Each patient’s clinical characteristics, including age, preoperative hematologic findings, and final biopsy results, were reviewed retrospectively using medical records. Based on CBC within 1 month before surgery, the specific hematologic findings analyzed were white blood cell (WBC) count, platelet count, neutrophil and lymphocyte counts, neutrophil and lymphocyte percentages, NLR, and PLR. Patients with pre-existing infections, a medical history of hematologic diseases, preoperative transfusion, other malignant diseases, and thrombolytic drugs were excluded as they may have had a confounding effect on the results of this study. Patients with tubo-ovarian abscesses or endometriosis were also excluded to exclude the effects caused by their respective inflammatory responses.

Data were analyzed using IBM SPSS statistics version V25.0 (IBM, Armonk, NY, USA) software and MedCalc Statistical Software version 19.4.0 software (MedCalc Software Ltd., Ostend, Belgium, 2020) was used for receiver operating characteristic (ROC) curve analysis. One-way analysis of variance was performed to compare the mean values of continuous variables, and post hoc analysis was performed using Scheffé testing procedure. Scheffé testing for post hoc analysis is generally used when the sample size in each group is unequal. Statistically significant differences were established between the groups when the *p* values were less than 0.05 at a confidence interval of 95%. An ROC curve analysis was performed to establish an appropriate cut-off level. We obtained a cut-off level that maximized Youden’s J statistic (sensitivity + specificity–1). The sensitivity, specificity, and area under the curve (AUC) were calculated, and binominal logistic regression was used to calculate odds ratios.

The current retrospective study was approved by the Institutional Ethics Committee of Daegu Catholic University Hospital (approval number: CR-20-209-L). All procedures in studies involving human participants were performed in accordance with the ethical standards of the institutional and national research committee and the 1964 Helsinki declaration and its later amendments or comparable ethical standards. The methodology used in the present study consisted of retrospective data collection; therefore, informed consent was not required.

## 3. Results

A total of 630 patients with ovarian tumors were enrolled in the present study. The final histopathological results revealed that 318 patients had benign ovarian tumors (epithelial [mucinous, serous, sero-mucinous, etc.]: *n* = 200; non-epithelial [mature cystic teratoma, fibroma, thecoma, etc.]: *n* = 118), 108 had borderline epithelial ovarian tumors, and 204 had malignant epithelial ovarian tumors.

The histopathology and characteristics of the enrolled patients with ovarian tumors are shown in Table 1. Differentiation grades and cancer stages were also specified for malignant ovarian tumors. A comparison of clinical characteristics and CBCs among benign, borderline, and malignant ovarian tumor groups is presented in Table 2.

The age range of each study group was significantly different because benign and borderline ovarian tumors occur at a relatively young age, while malignant ovarian tumors occur more often in older people. There were no statistically significant differences between NLRs and PLRs of patients with epithelial and non-epithelial benign ovarian tumors. White blood cell counts, hemoglobin densities, platelet counts, neutrophil counts, and lymphocyte counts were significantly different between patients with (1) benign or borderline ovarian tumors and (2) malignant ovarian tumors.

A comparison of the mean platelet values of each ovarian tumor is shown in Table 2 (benign (256,323.9 ± 66,984.9); borderline (245,027.8 ± 62,092.3); malignancy (280,828.4 ± 97,239.9), *p*-value (ANOVA test): *p* < 0.001, benign vs. borderline (*p* = 0.424); benign vs. malignancy [*p* = 0.002], borderline vs. malignancy (*p* = 0.001)). The NLRs of patients with malignant ovarian tumors were significantly higher than those of patients with benign or borderline ovarian tumors (benign (2.4 ± 2.2); borderline (2.7 ± 2.5); malignancy (3.9 ± 3.4)). The PLRs of patients with malignant ovarian tumors were also significantly higher than those of patients with benign or borderline ovarian tumors (benign (141.8 ± 62.0); borderline (146.9 ± 80.2); malignancy (194.8 ± 104.2)).

In sub-analysis of PLR and NLR with respect to stage, NLR in advanced ovarian cancer was statistically higher than in localized ovarian cancer (3.4 ± 3.1 (stage 1 and 2) vs. 4.4 ± 3.6 (stage 3 and 4), *p* = 0.043). In addition, PLR in advanced ovarian cancer was statistically higher than in localized ovarian cancer (171.3 ± 89.5 (stage 1 and 2) vs. 221.9 ± 113.5 (stage 3 and 4), *p* = 0.001). In addition, another sub-analysis to compare the mean and platelet–neutrophile ratio (PNR) values of each ovarian tumor was performed. (Benign (71.4 ± 30.1); borderline (64.3 ± 26.7); malignancy (63.0 ± 28.1), *p*-value (ANOVA test): *p* = 0.003, benign vs. borderline (*p* = 0.088); benign vs. malignancy (*p* = 0.005), borderline vs. malignancy (*p* = 0.931)).

The appropriate cut-off value, sensitivity, and specificity for differentiating between benign or borderline and malignant ovarian tumors using ROC curve analysis are shown in Figure 1 and Table 3. Based on our study result, the NLR and PLR were significantly higher in malignant than in benign or borderline ovarian tumors, and they did not differ significantly between benign and borderline ovarian tumors. Therefore, when analyzing the ROC curves, we performed the analysis with group benign or borderline versus malignant ovarian tumors. The appropriate cut-off value was determined to be the maximum value of Youden’s J statistic. The appropriate cut-off value of NLR (AUC = 0.692, *p* < 0.001) for differentiating between benign or borderline and malignant ovarian tumors was 2.36, with a sensitivity of 66.7% and specificity of 66.2% (Figure 1A). The appropriate cut-off value of PLR (AUC = 0.670, *p* < 0.001) for differentiating between benign or borderline and malignant ovarian tumors was 150.02, with a sensitivity of 58.8% and specificity of 66.9% (Figure 1B).

For clinical application, the cut-off was applied to NLR and PLR values by rounding the figures: when the NLR was 2.4 or higher, the odds ratio of malignant ovarian tumor was 3.796 (95% CI; 2.667–5.403); when the PLR was 150 or higher, the odds ratio of malignant ovarian tumor was 2.857 (95% CI; 2.026–4.030; Table 4).

## 4. Discussion

Ovarian cancer has the highest mortality rate among gynecological cancers; since most patients are asymptomatic in the early stages, ovarian cancer is usually found in the advanced stage [28]. Differential diagnosis between preoperative benign, borderline, and malignant ovarian tumors is primarily based on imaging tests, but it is often difficult. Tumor markers such as the CA125 and CA19-9 may also be elevated in other benign diseases other than cancer; furthermore, there are cases of ovarian cancer in which neither CA125 nor CA19-9 is elevated, which limits their roles in preoperative diagnosis.

In the present study, we attempted to distinguish between benign, borderline, and malignant ovarian tumors by analyzing CBC with differential count tests performed preoperatively for each patient. Inflammatory reactions contribute to the development and progression of tumor formation and oncogenesis [29,30]. Due to these inflammatory reactions, blood components such as platelets and neutrophils are recruited to the tumor microenvironment [31]. Compared to test results of patients with benign ovarian tumors, those of patients with malignant ovarian tumors revealed higher neutrophil counts and lower lymphocyte counts [32].

Thrombocytosis is often caused by reactive processes, such as acute infection, tissue damage, chronic inflammation, surgery, iron deficiency, rebound effect after bone marrow suppression, and malignancy. Although thrombocytosis is not a finding specific to malignancy, the association between platelet and cancer has been steadily increasing [5]. Regarding the association between cancer and thrombocytosis, various studies on the mechanism of interaction have also been reported. Cancer enhance hepatic thrombopoiesis, leading to increase platelet production in bone marrow. Production of thrombopoietic cytokine in liver and thrombocytosis are caused by interleukin-1 (IL-1), IL-3, IL-4, IL-11, and tumor-derived platelet factor 4 in tumor host tissues [33,34]. Granulocyte macrophage colony-stimulating factor (GM-CSF) and granulocyte colony-stimulating factor (G-CSF) also promote the production of platelets [35].

Several reports have indicated that NLRs and PLRs can be used as markers of systemic inflammatory responses [7,8,9,10,11,12,13,14,15]. NLRs and PLRs have been applied as useful biomarkers of diagnosis and prognosis in various types of malignancies [24,36,37,38]. Several studies have reported that the possibilities of malignancy and worse prognoses increase when the NLR and PLR increase [16,17,18,19].

Two studies analyzed both NLRs and PLRs in patients with borderline ovarian tumors. Polat et al. reported that NLRs and PLRs may help predict malignant, but not borderline ovarian tumors, even with microinvasive stromal invasion [13]. The study analyzed the average NLR (benign (3.1 ± 2.9); borderline (2.6 ± 1.5); malignancy (3.9 ± 3.8)) and PLR (benign (142.1 ± 55.7); borderline (148.1 ± 59.4); malignancy (191.9 ± 115.1)) for ovarian tumors. They found that the average NLR and PLR values in patients with malignancy were higher than those in patients with benign or borderline ovarian tumors. They reported that there was a statistically significant difference between the NLRs and PLRs of borderline and malignant ovarian tumors, and that the optimal cut-off values to predict ovarian malignancy using NLRs and PLRs were 2.47 (*p* = 0.02) and 144.3 (*p* = 0.05), respectively. They reported that their findings may be because borderline ovarian tumors do not accompany a systemic inflammatory response, even with microinvasion, unlike malignant tumors. Their findings are consistent with those of the present study.

Psomiadou et al. also reported the NLRs and PLRs in patients with benign, borderline, and malignant ovarian tumors (NLR: benign (2.3 ± 1.2); borderline (4.0 ± 2.7); malignancy (3.6 ± 2.7), and PLR: benign (134.6 ± 50.5); borderline (180.7 ± 88.0); malignancy (210.6 ± 98.6)) [24]. The NLRs and PLRs of borderline and malignant tumors were higher than those of benign ovarian tumors. They reported that there was no statistically significant difference between the NLRs and PLRs of patients with borderline tumors and those with malignant ovarian tumors. Their study results showed differences in this respect compared to the results of the present study. However, the Psomiadou et al. study analyzed a small sample size of patients with borderline ovarian tumors (*n* = 9) compared to the present study (*n* = 318).

The results of the present study showed that NLRs and PLRs were significantly elevated in blood samples from patients with malignant ovarian tumors compared to those from patients with benign or borderline ovarian tumors. Lymphocyte counts were significantly lower in patients with malignant tumors than those with borderline or benign ovarian tumors. Therefore, increased NLRs and PLRs indicate that a patient is more likely to have malignant ovarian tumors than benign or borderline ovarian tumors. The results of this study show that borderline ovarian tumors do not exhibit increased NLRs and PLRs compared to benign tumors, which may be because borderline ovarian tumors do not cause the systemic inflammation seen in patients with malignant tumors. This study showed that the NLR cut-off value for differentiating between benign or borderline and malignant ovarian tumors was 2.36, whereas the corresponding PLR cut-off value was 150.02. These results are similar to those of previous studies. Based on our study results, if the NLR is higher than 2.4 and/or PLR is higher than 150.0, there is a higher possibility of malignant ovarian tumors than benign or borderline tumors.

Several studies have reported poor prognosis in ovarian cancer patients with increased NLRs and PLRs compared to those with normal NLRs and PLRs. These studies have demonstrated that these parameters may indicate a poorer surgical outcome in patients with cancer [16,17,18,19]. Kokcu et al. reported that NLRs, PLRs, and platelet counts are independent prognostic factors for advanced-stage malignant ovarian masses [39]. In another study, platelet counts, NLRs, and PLRs were prognostic factors for progression-free survival (PFS) and overall survival (OS). Wang et al. reported that preoperative NLRs were a significant predictor of poor PFS and OS in malignant ovarian masses [40]. It is also reportedly associated with chemotherapy resistance [41].

In a sub-analysis of the results of this study, the NLRs and PLRs of patients with advanced stage (3 or 4) ovarian cancer were higher than those of patients with localized stage (1 or 2) ovarian cancer. Based on the results of previous studies and the current study, it is thought that NLRs and PLRs increase as cancer progresses. Therefore, increases in NLRs and PLRs are associated with advanced ovarian cancer and may be associated with poor prognosis. In addition, results from the PNR sub-analysis show that both thrombocytosis and/or increase of neutrophile can be present in malignant tumors, but in relative terms thrombocytosis is more pronounced in malignant tumors.

NLR and PLR increases are not cancer-specific findings. In addition, an increase in these values is not indicative of an absolute risk of ovarian cancer and may be transient depending on a variety of conditions. However, CBC with a differential count is a common and inexpensive preoperative test. Therefore, even if it is not a confirmatory test for ovarian cancer, it has clinical utility as a potential auxiliary tool for differential diagnosis before surgery. The exact diagnostic cut-off values for NLRs and PLRs for diagnosing malignant ovarian tumors have not yet been established. Based on the results of previous studies and this study, the cutoff level for discriminating between malignant tumors and benign or borderline tumors is estimated to be approximately 2.5 for NLRs and 150 for PLRs.

One limitation of this study is that it was retrospective and not a large-scale study. Two strengths of this study are that it (1) included a larger number of patients with borderline ovarian tumors compared to that in previous studies, and (2) the number of patients was allocated proportionally in consideration of the prevalence of benign, borderline, and malignant ovarian tumors.

## 5. Conclusions

In conclusion, the NLRs and PLRs of malignant tumors were significantly higher than those in benign or borderline ovarian tumors, and the NLRs and PLRs between benign and borderline ovarian tumors did not differ significantly. A high preoperative NLR and PLR mean that the likelihood of epithelial ovarian cancer is higher than that of benign or borderline tumors.

## Figures and Tables

**Figure 1 jcm-11-01355-f001:**
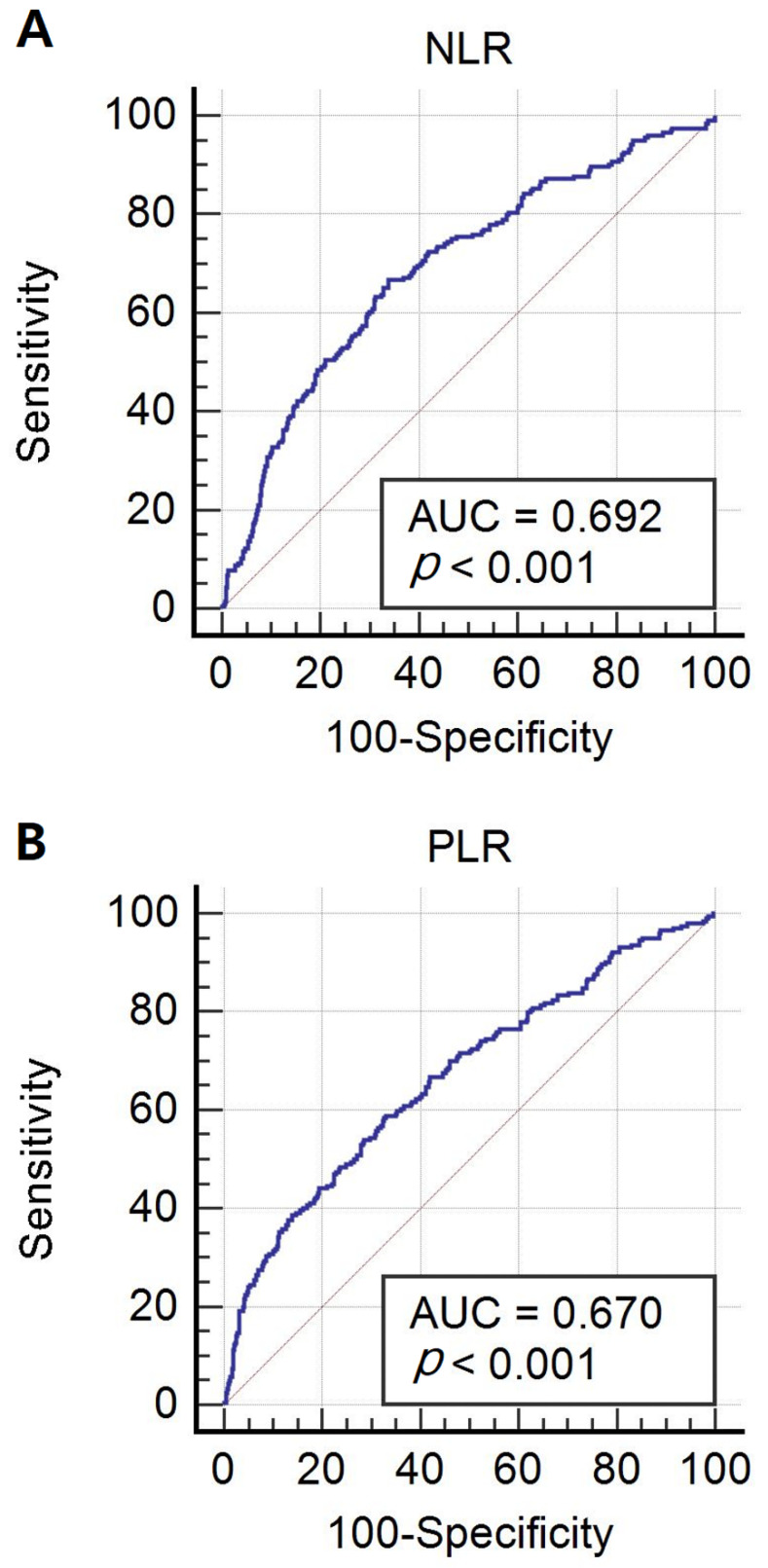
Receiver operating characteristic curve analysis of NLR (**A**) and PLR (**B**) in patients with benign or borderline versus malignant ovarian tumor. Abbreviations: NLR, neutrophil to lymphocyte ratio; PLR, platelet to lymphocyte ratio; AUC, area under curve.

**Table 1 jcm-11-01355-t001:** Histopathology and characteristics of the enrolled patients with ovarian tumor.

Characteristic	Number of Patients
Benign ovarian tumor (*n* = 318)	
Histopathology Epithelial ovarian tumor (*n* = 200)	
Mucinous cystadenoma	100 (31.4%)
Serous cystadenoma	77 (24.2%)
Sero-mucinous cystadenoma	15 (4.7%)
Mucinous cystadenofibroma	2 (0.6%)
Serous cystadenofibroma	3 (0.9%)
Sero-mucinous cystadenofibroma	3 (0.9%)
Non-epithelial ovarian tumor (*n* = 118)	
Mature cystic teratoma	99 (31.1%)
Fibroma	10 (3.1%)
Fibrothecoma	7 (2.2%)
Thecoma	1 (0.3%)
Sclerosing stromal tumor	1 (0.3%)
Borderline ovarian tumor (*n* = 108)	
Histopathology	
Mucinous borderline tumor	83 (76.9%)
Serous borderline tumor	19 (17.6%)
Sero-mucinous borderline tumor	4 (3.7%)
Endometrioid borderline tumor	2 (1.9%)
Malignant ovarian tumor (*n* = 204)	
Histopathology	
High-grade serous carcinoma	71 (34.8%)
Endometrioid adenocarcinoma	44 (21.6%)
Mucinous adenocarcinoma	40 (19.6%)
Clear cell carcinoma	25 (12.3%)
Mixed epithelial carcinoma	8 (3.9%)
Low-grade serous carcinoma	7 (3.4%)
Carcinosarcoma	5 (2.5%)
Undifferentiated carcinoma	3 (1.5%)
Sero-mucinous adenocarcinoma	1 (0.5%)
Differentiation grade	
Grade 1 (well diff.)	29 (14.2%)
Grade 2 (moderately diff.)	93 (45.6%)
Grade 3 (poorly diff.)	82 (40.2%)
Stage	
Stage I	
IA	45 (22.1%)
IB	5 (2.5%)
IC	35 (17.2%)
Stage II	
IIA	4 (2.0%)
IIB	11 (5.4%)
IIC	9 (4.4%)
Stage III	
IIIA	3 (1.5%)
IIIB	9 (4.4%)
IIIC	68 (33.3%)
Stage IV	15 (7.4%)

Abbreviations: diff.; differentiation.

**Table 2 jcm-11-01355-t002:** Comparison of clinical characteristics and complete blood count among study groups.

	Pathology	Mean ± SD	*p*-Value (ANOVA)	Comparison between Groups ^a^	*p*-Value (Post Hoc ^b^)
Age (*n* = 630)	Benign (*n* = 318)	45.3 ± 16.5	*p* < 0.001	1 vs. 2	*p* = 0.487
	Borderline (*n* = 108)	47.3 ± 17.2		1 vs. 3	*p* < 0.001
	Malignant (*n* = 204)	52.9 ± 12.0		2 vs. 3	*p* = 0.010
White blood cell (/µL)	Benign	6831.4 ± 2529.2	*p* = 0.007	1 vs. 2	*p* = 0.966
	Borderline	6760.2 ± 1946.2		1 vs. 3	*p* = 0.014
	Malignant	7470.6 ± 2517.8		2 vs. 3	*p* = 0.050
Hemoglobin (g/dL)	Benign	12.7 ± 1.3	*p* < 0.001	1 vs. 2	*p* = 0.963
	Borderline	12.8 ± 1.3		1 vs. 3	*p* < 0.001
	Malignant	12.0 ± 1.4		2 vs. 3	*p* < 0.001
Platelet count (/µL)	Benign	256,323.9 ± 66,984.9	*p* < 0.001	1 vs. 2	*p* = 0.424
	Borderline	245,027.8 ± 62,092.3		1 vs. 3	*p* = 0.002
	Malignant	280,828.4 ± 97,239.9		2 vs. 3	*p* = 0.001
Neutrophil count (/µL)	Benign	4192.8 ± 2319.7	*p* < 0.001	1 vs. 2	*p* = 0.886
	Borderline	4317.9 ± 1753.7		1 vs. 3	*p* < 0.001
	Malignant	5156.5 ± 2464.1		2 vs. 3	*p* = 0.009
Lymphocyte count (/µL)	Benign	1992.0 ± 647.0	*p* < 0.001	1 vs. 2	*p* = 0.227
	Borderline	1868.5 ± 615.2		1 vs. 3	*p* < 0.001
	Malignant	1660.5 ± 651.2		2 vs. 3	*p* = 0.025
NLR	Benign	2.4 ± 2.2	*p* < 0.001	1 vs. 2	*p* = 0.648
	Borderline	2.7 ± 2.5		1 vs. 3	*p* < 0.001
	Malignant	3.9 ± 3.4		2 vs. 3	*p* = 0.002
PLR	Benign	141.8 ± 62.0	*p* < 0.001	1 vs. 2	*p* = 0.850
	Borderline	146.9 ± 80.2		1 vs. 3	*p* < 0.001
	Malignant	194.8 ± 104.2		2 vs. 3	*p* < 0.001

Abbreviations; SD: Standard deviation, ANOVA: Analysis of variance, vs.: versus, NLR: Neutrophil to lymphocyte ratio, PLR: Platelet to lymphocyte ratio ^a^ group 1: Benign ovarian tumor, group 2: Borderline ovarian tumor, group 3: Malignant ovarian tumor. ^b^ Post Hoc analysis: A Scheffé test was used.

**Table 3 jcm-11-01355-t003:** Appropriate cut-off value, sensitivity and specificity for differentiating benign/borderline and malignant ovarian tumor using ROC curve analysis.

	Cut-Off	Sensitivity (%)	Specificity (%)
NLR Benign or borderline vs. malignancy	2.36	66.7	66.2
PLR Benign or borderline vs. malignancy	150.02	58.8	66.9

Abbreviations; ROC: Receiver operating characteristic, NLR: Neutrophil to lymphocyte ratio, PLR: Platelet to lymphocyte ratio, vs.: versus.

**Table 4 jcm-11-01355-t004:** Odds ratio of malignant ovarian tumors according to NLR and PLR.

	Odds Ratio ^a^	95% CI	*p*-Value
NLR ≥ 2.4 Malignancy vs. Benign or borderline	3.796	2.667–5.403	*p* < 0.001
PLR ≥ 150.0 Malignancy vs. Benign or borderline	2.857	2.026–4.030	*p* < 0.001

Abbreviations; NLR: Neutrophil to lymphocyte ratio, PLR: Platelet to lymphocyte ratio, vs.: versus, CI: Confidence interval ^a^ Binominal logistic regression was done.

## Data Availability

Data are available from the corresponding authors upon reasonable requests.

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
