# Peer review of "Neutrophil–Lymphocyte and Platelet–Lymphocyte Ratios in Preoperative Differential Diagnosis of Benign, Borderline, and Malignant Ovarian Tumors"

_jcm, 2022, doi:10.3390/jcm11051355_

Round 1
Reviewer 1 Report
Excellent paper. Well written and important.
This is a cross sectional study to evaluate parameters in the CBC to predict he phenotype of ovarian tumour according to degree of invasiveness
This is a confirmatory study but the field is understudied and the importance has not been adequately appreciated
Opens a lot of questions about the biology behind the findings
Should state that the ratios are not risk factors but are markers of tumour burden. They are transient
Do the markers decline after surgery/chemotherapy
Would like to see some data on platelets alone
See recent paper by Giannakeas et al in jama network open
Would like to compare platelets alone to platelet/neutrophil ratio.
Would expand the discussion into possible mechanism
I would remove the abbreviations and spell out the ratios in full words
Author Response
Please read the attached file.
Sincerely,
Jung Min Ryu.

Reviewer 2 Report
This manuscript on using NLR and PLR in a pre-operative setting to improve assessment of ovarian masses is an interesting original research manuscript. There needs to be a major revision before any consideration for publication.
Major concerns:
Methodology:
- there is no identification during which time period, these samples were analysed (how many years)
- how was the sample size determined? In line 72-74 just a random number selection is presented. What is the rationalle for this number?
- describe the standard process of determining the NLR and PLR - so technical laboratory approach, how is this calculated? Is it just divided any other formula?
- why do the authors not compare the traditional pre-op diagnostics (inclusive of CA 125 and CA 19-9, imaging diagnostics) and add NLR and PLR additionally? In a real-life setting, this will happen in concordance and at least an adjustment for the tumour markers should be available
- there is no description of the FIGO staging used to determine the tumours
- were all tumours pre-operatively persumed to be benign/malignant/borderline? Add this information - the persumed type pre-operatively and correlate it with final outcome and also PLR and NLR
Results:
- It is highly unsual to not report the statistically insignificant data - please report also that data (Table 2)
- Please structure the results so the reader can better understand what the ROC was for? Did you group together benign vs. borderline plus ovarian
Minor concerns:
- while the summary does not need to be structured some justification on why you are doing this as well as a better methodological description is needed
- discussion lines 170 - 175 would fit better in the intro, that needs to be redone. More justification and explanation needs to be given why you decided to do this and what the mechanism of action is
- focus in the discussion on your results in connection with topics of pre-operative discrimination; what has been tried? what does this add?
Author Response

(The authors gave the same response as above.)
